# Analyzing the efficacy of trimethylolpropane trioleate oil for predicting cutting power and surface roughness in high-speed drilling of Al-6061 through machine learning

Pramod S. Kathmore[1], Bhanudas D. Bachchhav[2]*, Duran Kaya[3], Sachin Salunkhe[4,5], Lenka Cepova[6], Ondřej Mizera[6], Emad Abouel Nasr[7]

1 Department of Technology, Savitribai Phule Pune University, Pune, India, 2 Department of Mechanical Engineering, All India Shri Shivaji Memorial Society's, College of Engineering, Pune, India, 3 Gazi University Project Coordination and Implementation Research Center, Ankara, Türkiye, 4 Department of Biosciences, Saveetha School of Engineering, Saveetha Institute of Medical and Technical Sciences, Chennai, India, 5 Department of Mechanical Engineering, Gazi University Faculty of Engineering, Maltepe, Ankara, Türkiye, 6 Department of Machining, Assembly and Engineering Metrology, Faculty of Mechanical Engineering, VSB —Technical University of Ostrava, Ostrava, Czech Republic, 7 Department of Industrial Engineering, College of Engineering, King Saud University, Riyadh, Saudi Arabia

* bdbachchhav@aissmscoe.com

**Data Availability Statement:** All relevant data are within the manuscript.

## Abstract

This study aimed to investigate the impact of a lubricant derived from trimethylolpropane trioleate on power consumption and surface roughness during high-speed drilling of Al-6061, with the goal of developing an environmentally friendly cutting fluid. The study investigated the impact of additive concentration, spindle speed, and feed rate on energy consumption and surface roughness using a Taguchi L27 orthogonal array. Through analysis of the Taguchi experimental outcomes and single-to-noise ratios, the parameters were ranked accordingly. The results of the ANOVA analysis reveal that spindle speed has the greatest impact on Power (87.89%), followed by followed feed rate (6.96%) and additive concentration (2.98%). However, feed rate (43.51%) has the most significant influence on surface roughness, followed by speed (38.48%) and additive concentration (11.90%). Varying additive concentration affects more on surface quality rather than power consumption. Furthermore, a machine learning algorithm was developed to forecast and compare various key aspects of high-speed drilling machinability, including power and surface roughness. Three different measures of accuracy were used to evaluate the performance of the projected values: coefficient of determination (R2), mean absolute percentage error, and mean square error. The decision tree performed better than other models in accurately predicting power and surface roughness. This research introduces an innovative method for assessing the most effective biodegradable cutting fluid and forecasting power and surface quality by developing an optimal combination.

**Funding:** The authors would like to express their gratitude to King Saud University, Riyadh, Saudi Arabia, for sponsoring this work under Researchers Supporting Project number (RSP2024R164). This article was co-funded by the European Union under the REFRESH – Research Excellence For Region Sustainability and High-tech Industries project number CZ.10.03.01/00/22_003/0000048 via the Operational Programme Just Transition and has been done in connection with project Students Grant Competition SP2024/087 „Specific Research of Sustainable Manufacturing Technologies" financed by the Ministry of Education, Youth and Sports and Faculty of Mechanical Engineering VŠB-TUO. Article has been done in connection with project Students Grant Competition SP2024/087 „Specific Research of Sustainable Manufacturing Technologies" financed by the Ministry of Education, Youth and Sports and Faculty of Mechanical Engineering VŠB-TUO. The testing samples are manufactured at the funder institute. Furthermore, the funder also carries out the test and helps with the development of the machine learning model. In addition, the funder revised the manuscript per the model and compared the results with the ML model.

**Competing interests:** The authors have declared that no competing interests exist.

## 1. Introduction

Friction is crucial in machining, influencing cutting force intensity and, consequently, power consumption. When a tool engages a work-piece, friction between the tool and material impedes motion, generating cutting forces. Overcoming this resistance necessitates higher cutting forces to maintain the desired cutting action, thereby increasing power consumption. Lubricating oil plays a significant role in mitigating friction and optimizing machining efficiency. Cutting fluid serves the purpose of dissipating heat generated during metal cutting while reducing friction between the tool and work-piece interface, thus enabling smoother and more efficient cutting.

Globally, there is a widespread prohibition on the use of hazardous mineral-based lubricating oils, largely because of their substantial environmental hazards. Mineral based lube hardly suffice the need of cutting fluids without the addition of costly additives, making them less feasible for industrial applications. In contrast, vegetable-based lubricants offer an eco-friendly alternative with notable advantages such as high lubricity, polarity, a high viscosity index, a high flash point, and a low evaporation rate. However, their full adoption hasn't been realized due to concerns about oxidation stability and unsuitable chemical structures. Fortunately, chemical modifications of vegetable oils address these structural issues, rendering them suitable replacements for cutting fluids. Several review explores both advancements and challenges in formulating vegetable-based lubricants, focusing on their suitability for metal cutting operations [1]. In comparison to vegetable oils, mineral-based metal cutting fluids exhibit lower lubricity. Nonetheless, they pose significant drawbacks as they are toxic, non-biodegradable, and harmful to human health. An estimated 16% of the manufacturing cost is attributed to the handling, maintenance, and disposal of cutting fluid [2–5]. Various multi-attribute decision-making techniques are utilized to categorize vegetable-based lubricants and choose the most suitable lubricant for particular metal-cutting applications [6, 7]. Researchers are investigating trimethylolpropane trioleate (TMPTO) as a potential environmentally friendly lubricant additive and assessing its effectiveness under various operational conditions. TMPTO is being considered for its potential as a bio-lubricant due to its low toxicity and excellent biodegradability, crucial properties for mitigating the environmental impact of lubricants. Furthermore, the addition of chemically sulfurized vegetable oil-based additives enhances the lubricating capabilities of TMPTO base oil. These additives are recognized for their outstanding anti-wear and extreme pressure (EP) properties, forming a protective film on metal surfaces. This film acts as a barrier, reducing friction and wear, particularly under high-pressure conditions, thereby preventing direct metal-to-metal contact and minimizing frictional losses. Bio-lubricants derived from vegetable oils, modified chemically with TMP, offer a sustainable and environmentally friendly alternative to mineral-based lubricating oils. They exhibit similar performance characteristics and find applications across diverse industries, contributing to the reduction of overall environmental impact and advocating for a more sustainable approach to lubrication [8–11].

The cooling of metal cutting processes is rather more important than the lubrication; however, this does not apply to high-speed cutting of non-ferrous metals because of their high ductility. In high-speed machining of non-ferrous metals, lubrication is essential for cooling and lubricating the tool and work-piece surfaces. In addition, lubrication increases tool life and improves surface quality by reducing friction. Additionally, lubrication helps to reduce tool wear and work-piece deformation. In ductile materials such as aluminum alloy, high-speed cutting can lead to rapid transfer of cutting heat to the workpiece, leading to chip adhesion to the tools, consequently causing diminished cutting ability, machining precision, and surface quality due to chip accumulation, thereby obstructing the cutting area and intensifying the cutting process difficulty, necessitating constant cooling and lubrication by cutting fluid, which

cannot effectively remove the cut debris [12, 13]. The effects of a trimethylolpropane trioleate (TMPTO)-based lubricant on thrust force and torque during high-speed drilling of Al-6061 were investigated as an effective environmentally friendly cutting fluid. A machine learning model was also developed to predict and compare several significant aspects of high-speed drilling machinability, including thrust force and torque [14]. Introduction of graphene and boron hexagonal nitride nanoparticles resulted in improved tribological performances in metal cutting applications [15, 16]. In 2015, Wu et al. endeavored to optimize the synthesis process of trimethylolpropane trioleate (TMPTO) by utilizing oleic acid (OA) and trimethylol-propane [17]. Polyol ester-based vegetable oils demonstrate promising properties in comparison to mineral-based lubricants when utilized in high-speed drilling of aluminum and stainless steel materials. Introducing graphene-based nanofluid yields a notable reduction in drilling torque, approximately 25% lower than that of a standard emulsion lubricant. Consequently, employing this nanofluid leads to a substantial increase in tool lifespan, up to 20 times longer than when using a conventional emulsion lubricant [18]. Polyol ester-derived vegetable oils demonstrate advantageous behaviors when compared to mineral-based lubricants in high-speed drilling operations in aluminum and stainless steel materials [10, 11, 14, 19–21].

Machine learning (ML) has been extensively applied in metal cutting processes for predicting cutting forces, monitoring tool conditions, optimizing processes, ensuring quality control, and more. A variety of methods are employed to forecast cutting forces and torque in drilling, including artificial neural networks [22], response surface methodology [23–25], and the utilization of diverse machine learning algorithms [26–28], both under dry and lubricated conditions.

In this study an attempt have been to evaluate lubricant efficacy in high speed drilling of Al-6061 in order to save energy and achieve better surface quality. An effect of cutting speed and feed rate and additive concentration in TMPTO based oil were tested to determine desired cutting power and surface roughness using predictive modeling methods such as linear regression, random forests, decision trees, and support vector machines. The accuracy of these models was assessed through metrics like root coefficient of determination ($R2$), mean square error and mean absolute error.

## 2. Materials and sample preparation

### 2.1 Lubricants

TMPTO was chosen as the base oil for machining due to its outstanding performance and eco-friendliness. It provides stable viscosity across temperatures, excellent low-temperature fluidity, and a high flash point for safety. With strong hydrolytic and foaming stability, it ensures prolonged use while enhancing tool performance and reducing wear. As a biodegradable alternative to mineral oils, TMPTO is a sustainable choice for machining. TMPTO (trimethylolpropane trioleate) was selected as the base oil because of its notable rheological properties and high content of long-chain fatty acids. TMPTO offers favorable characteristics which are particularly suitable for the cutting fluids [10, 11, 14, 29, 30]. There are three additives that were procured from the local market in order to make the blend. The blends were uniformly mixed using a magnetic stirrer. The formulated samples contains chemically modified sulfurized vegetable oil olefin additive at 4%, 6% and 8% concentration in TMPTO base oil. For preliminary screening of the performance of formulated lube oil, an anti-wear assessment was conducted in accordance with the ASTM D4172 standard, while an extreme pressure evaluation was performed following the ASTM D2783 standard using four ball tester. The AWSD ranged from 0.37 mm to 0.44 mm for additive concentrations between 4% and 6%. EP loads of 2000 N,

6082 N, and 7848 N were observed for additive concentrations of 4%, 6%, and 8%, respectively. Table 1 shows the rheological properties of samples prepared.

Infrared spectroscopy was used to characterize the molecular structure of the formulated TMPTO base oil as shown in Fig 1. There was a vibration peak observed for Olefinic (alkane) C-H stretching a distance of 3,004.38 cm and bending a distance of 1,461.37 cm to 1,354.68 cm. There is a peak at the distance of 1,740.90 cm for the stretching vibration of TMPTO, and a peak at the distance of 1,461.37 cm for the bending vibration of TMPTO. There is also a peak at 721.33 cm that present in the extended alkyl chain of TMPTO. The peaks in the spectra of TMPTO and A bending and stretching motion of olefinic (alkane) C-H has been observed and used to determine the structure of the compound. This data was then used to confirm the structure and the properties of TMPTO.

## 2.2 Equipment

In metal cutting, cooling is often prioritized over lubrication. However, in high-speed cutting of ductile, non-ferrous metals like Al6061 aluminum alloys, lubrication becomes essential. It reduces friction, aids cooling, and prevents chip buildup, which can impair cutting performance, accuracy, and surface finish. Cutting fluids are crucial for both cooling and lubricating, ensuring optimal conditions and tool performance by removing debris effectively.

This study sought to explore and examine the effect of additive level, spindle speed and feed rate on Power applied during high-speed drilling, as well as the quality of the finished surface on Al6061. The drilling operation was conducted on a 152 mm X 152 mm X 20 mm Al-6061 plate. Specific details about the material's chemical composition and mechanical properties are presented in Tables 2 and 3 respectively. An HSS drill bit with an 8 mm diameter was utilized for the drilling process across three different spindle speeds, feed rates, and concentrations of additives respectively. Statistical analysis techniques were employed to evaluate the results and determine the most favorable combination of feed rate, spindle speed, and additive concentration for optimal drilling performance.

A high-speed drilling operations were conducted using a BFW Chakra 60 CNC Machining Center at R.K. Engineering Pvt Ltd, Pune. For the purpose of monitoring and recording power and surface roughness, a wireless spike tool holder dynamometer was used. The data was collected using data acquisition systems so that further modeling and analysis could be performed. In order to measure the values and qualitative characteristics of the roughness of the surface, a surface roughness tester (Mitutoyo CS5000) was deployed. Fig 2 depicts an experimental set-up.

## 2.3 Experimental design

The purpose of this study was to investigate how additive concentration, feed rate, and spindle speed affect power and surface roughness. In order to calculate the average required power

**Table 1. Rheological properties of oil samples.**

| Base Oil | TMPTO |
|---|---|
| Additives | Chemically modified sulfurized vegetable oil olefins @ 4%, 6%, 8% |
| Acid Value (mg KOH/mg) | 0.1 |
| Density (15˚C g/cm$^3$) | 1 |
| Viscosity (40 mm$^2$/s) | 40 |
| Pour Point (˚C) | -15 |
| Flash Point (COC,˚C) | 164 |

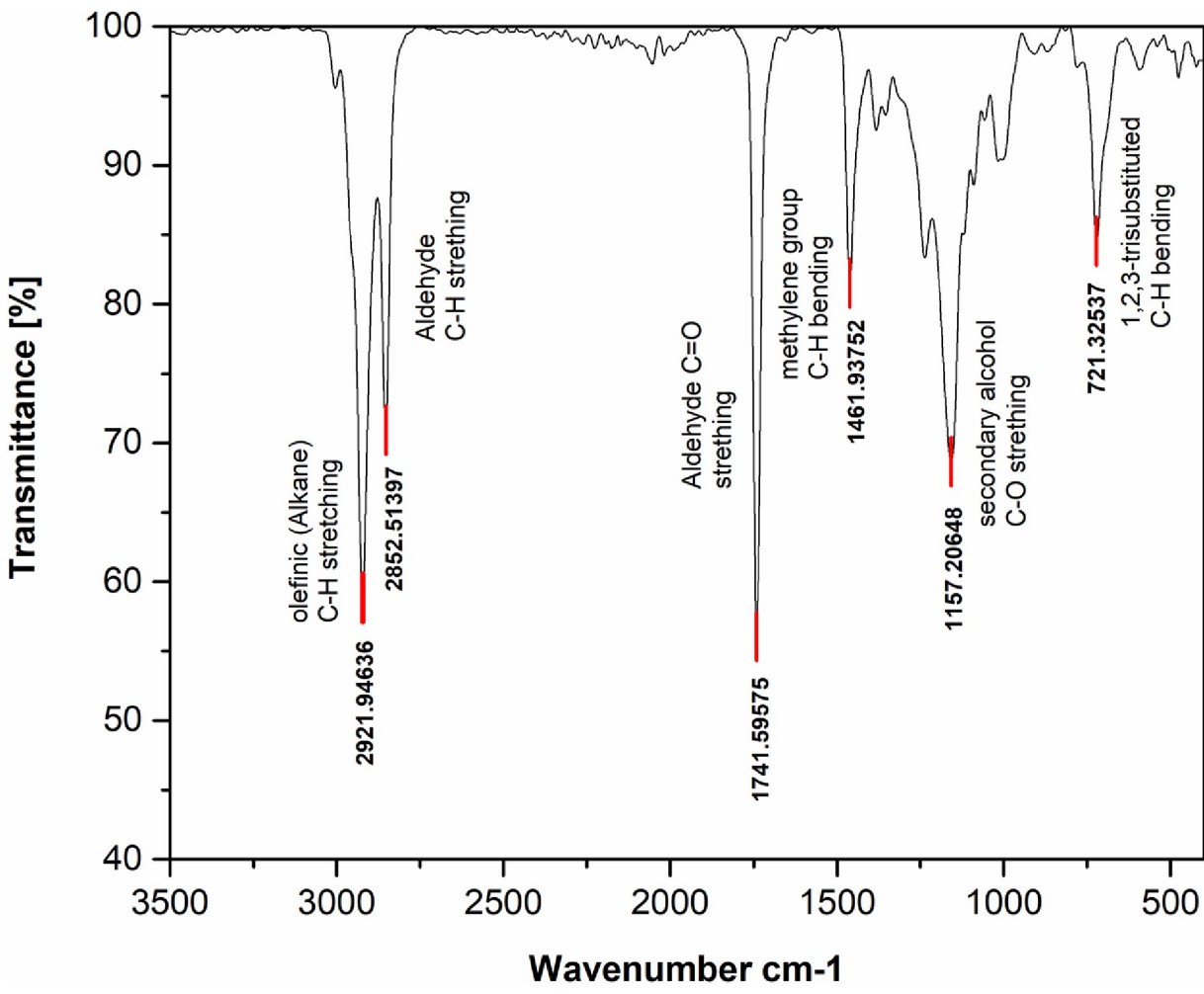

**Fig 1. FTIR analysis of the biolubricant derived from trimethylolpropane trioleate oil.**

(P), the Eq (1) used.

$$P = T \times w \tag{1}$$

Where, P indicated Power (kW), T indicate the torque (Nm) and w is spindle speed (RPM). Torque measurements were captured using a wireless spike tool holder dynamometer. The effect of additive concentration in actual contact conditions was evaluated in this experiment by altering the additive concentration in the best sample oil based on the anti-wear performance. To assess the efficiency of drilling, feed rate and speed are the two key factors that have the most influence. It is also important to consider drill size. However, the size of the drill is impacted by the speed of the spindle. Spindle speed affects the speed and accuracy of drilling, as well as the size of the drill bit. In this study, only one drill size set was examined. The effect of additive concentration in actual contact conditions were evaluated in this experiment by

**Table 2. Composition of Al6061 in chemical terms.**

| Components | Cu. | Fe. | Mn. | Mg. | Cr. | Si. | Others | Al |
|---|---|---|---|---|---|---|---|---|
| Amount (%) | 0.1 | 0.5 | 0.5 | 0.8 | 0.25 | 0.9 | 0.3 | Others |

**Table 3. A 6061 alloy's mechanical properties.**

| Material | Coefficient of Thermal Expansion K-1 | Thermal Conductivity W/m.K | Elastic Modulus GPa | Electricity Resistivity Ὼ.m | Melting Point (OC) |
|---|---|---|---|---|---|
| | | | | | **Min** |
| | | | | | **Max** |
| Al 6061 | $24 \times 10^{-6}$ | 180 | 70 | $0.038 \times 10^{-6}$ | 555 |
| | | | | | 650 |

altering the additive concentration. A **Table 4** provides an overview of the three levels and three key controlling factors that were considered during the investigation.

In the proposed study, Taguchi's orthogonal array proposes conducting 27 experiments using the $L_{27}$ orthogonal array. The experimental designs were thus derived from the $L_{27}$ orthogonal array. Every response factor was duplicated twice to assess variance and S/N ratios of the outcomes, resulting in 54 observations in total. After completing the experiment, each response parameter is converted to an S/N ratio. In this research, we compared the calculated signal-to-noise ratio values using the acquired data to evaluate any discrepancies or similarities. The function of smaller is better indicated in **Eq (2)** is considered an objective function since all response parameters are expected to be smaller in this study.

$$\frac{S}{N} = -\log\frac{1}{n}\left(\sum\nolimits_{i=1}^{n} y^2\right) \tag{2}$$

## 2.4 Predictive models in machine learning

Machine learning models rely on several key assumptions, including: independent observations, identically distributed data, linearity (for linear models), relevant features, no multicollinearity, normally distributed errors (in some models), model simplicity, sufficient data, and handling noisy data. These assumptions are considered for proper model performance and interpretation.

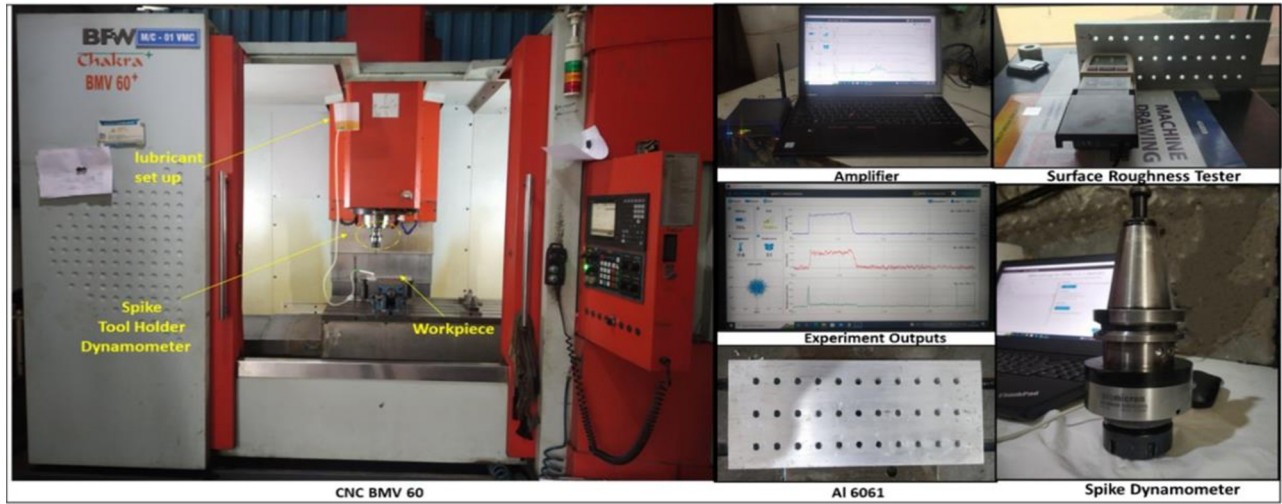

**Fig 2. Experimentation setup.**

**Table 4. Parameters and levels of the process.**

| Contributing factors | Levels | | | Unit |
|---|---|---|---|---|
| | 1 | 2 | 3 | |
| Concentration (A) | 4 | 6 | 8 | % |
| Spindle Speed (B) | 3053 | 4580 | 6107 | RPM |
| Feed Rate (C) | 0.1 | 0.2 | 0.3 | Mm/rev |

**2.4.1 Multiple linear regression.** The Multiple Linear Regression is a statistical method used in datasets where there is one dependent variable and multiple independent variables. This type of regression shows a linear connection between the dependent variable and two or more independent variables, hence its name as linear regression. In this approach, the dependent variable (y) is predicted based on several independent variables (x1, x2, x3 . . . xn), as represented in Eq (3).

$$y = w_1 * x_1 + w_2 * x_2 + w_3 * x_3 + \cdots + w_n * x_n + b \tag{3}$$

where,

$x_1, x_2, x3, \ldots, xn$ are the independent variables,

$y$ is the dependent variable,

$n$ is a positive integer.

$b$ is the bias or the intercept, and

$w_1, w_2, w_3, \ldots, w_n$ are the weights,

Linear regression uses the least-squares method to find the best-fit line for a given set of data. This involves calculating the slope and y-intercept that minimize the squared distance between the actual value and the predicted value as depicted in **Fig 3**

There are several factors that need to be considered when calculating the accuracy of a linear regression model, including the quality and quantity of the data, the independent variables, and the assumptions made about how the variables are related. The more accurate the assumptions, the better the model will perform. Additionally, the linear regression model should be tested on unseen data to ensure it is not overfitting the training data. Finally, the model should be evaluated using appropriate metrics.

**2.4.2 Decision tree.** At its simplest level, it can be described as a visual diagram that illustrates the clear pathway to making a decision. Fig 4 presents the visual depiction of a Decision Tree.

The decision tree is divided into three parts, as depicted in **Fig 4**, which contains three nodes: a root node, an interior node, and a leaf node [18]. The root node is the first node in the tree and it corresponds to the whole dataset. The internal nodes signify a single feature and the leaf nodes signify the response classes. Every branch of the tree signifies a judgment that is made based on the feature values. The decision tree is built by splitting the root node into smaller sub-trees until all the leaves are pure. This is done by repeatedly selecting the best split point based on a certain criterion. The tree is then pruned to remove any unnecessary splits and improve the accuracy of the model. The final tree is then used to make predictions on unseen data and the accuracy is assessed. The goal of the decision tree is to create a model that has high accuracy and is easy to interpret and visualize.

CART is a decision tree algorithm associated with regression tasks, and it belongs to a family of four decision trees, including Classification and Regression Trees, Iterative Dichotomiser, Reduction in Variance and Chi-Square. It involves constructing a decision tree model from a provided training dataset, making it an effective technique for both classification and

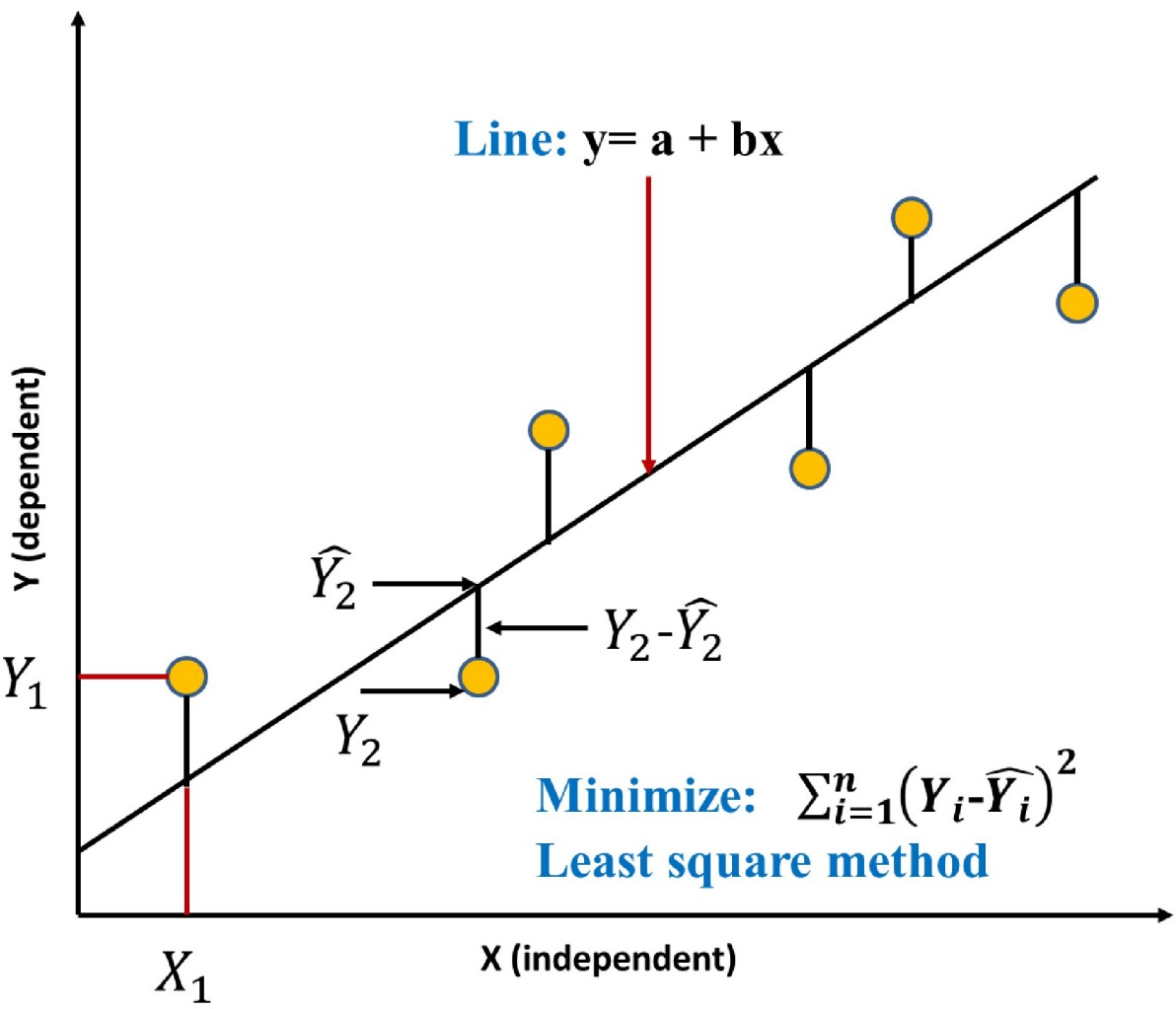

**Fig 3. Linear regression curve fitting to data points.**

regression issues. The trees generated by CART can effectively illustrate how a specific dataset is divided based on certain criteria.

In CART regression, the data is divided into two groups by identifying the threshold that produces the smallest total square residuals. This procedure is then repeated until a stopping condition is met for further division. If no stopping criterion exists, the model will perfectly fit the data, indicating potential overfitting and a likelihood of poor performance with new data. Such a model may exhibit low bias but high variance. To prevent overfitting, mitigation techniques can be employed such as splitting observations when there are only a minimum number of samples remaining or setting a maximum depth for the tree. In building decision trees with CART for classification scenarios, Gini impurity is used while least squares are utilized in regression cases to minimize residual sum of squares between observations and mean in each node (Mathematically expressed as Eq 4).

$$\varepsilon_i = Y_i - \hat{Y}_i \tag{4}$$

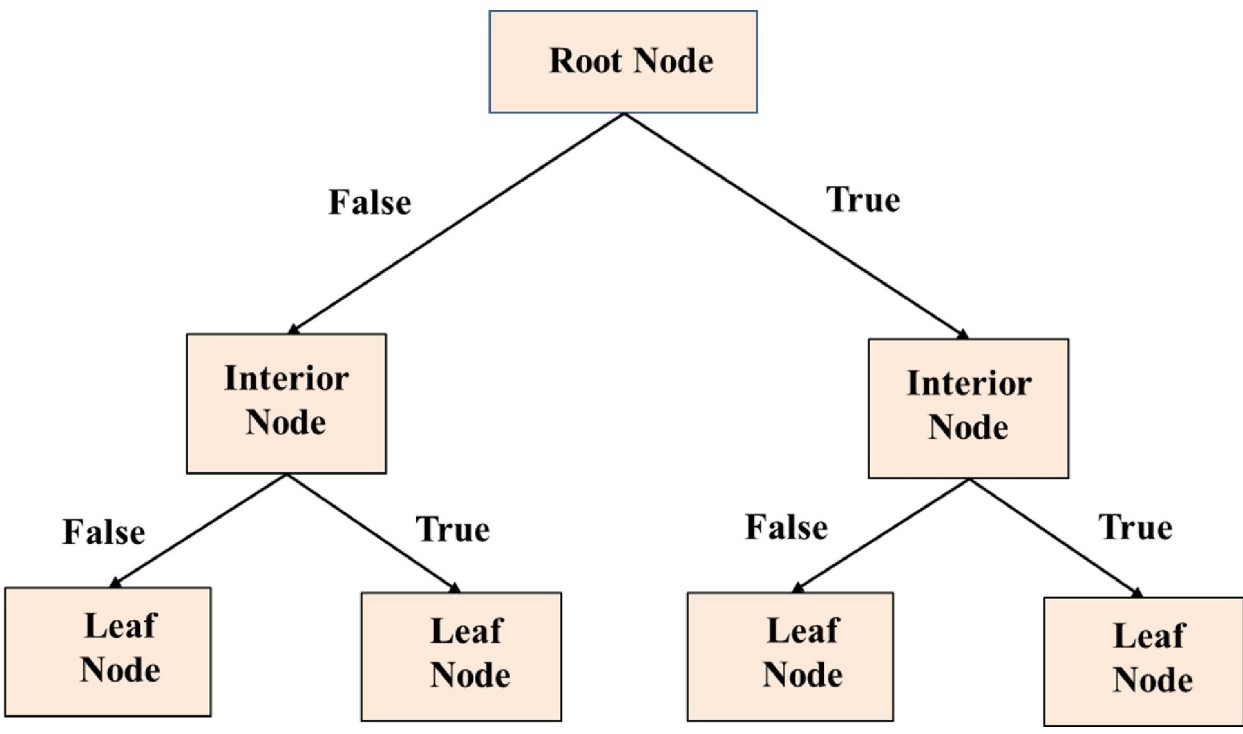

**Fig 4. Decision tree graphical illustrations.**

In terms of mathematics, we can write RSS (residual sum of squares) as **Eq 5**

$$RSS = \sum_{i=1}^{N} \left( Y_i - \hat{Y}_i \right)^2$$

$$RSS = \varepsilon_1^2 + \varepsilon_2^2 + \cdots + \varepsilon_n^2 \tag{5}$$

**2.4.3 Random forest regression.** As a machine learning technique, Random Forest Regression approach that entails training numerous random decision trees on a portion of the dataset in order to construct a model, resulting in increased stability and reduced variance. This approach is widely favored for its effectiveness with large and diverse datasets in regression analysis. The algorithm creates individual trees using distinct input data to inform the splitting of features at each node, selecting these features randomly. Additionally, independently trees run without interaction between them. To generate a individual forecast, the forecasts made by all the trees are combined to generate an average result known as the Random Forest prediction. In classification problems, the ultimate output is determined by calculating the mean of all outputs. Similarly, for regression problems, it entails computing the mean of all outputs. This part of the process is referred to as Aggregation. A graphical representation displayed in Fig 5 depicting a random forest. In classification and regression tasks, predetermined forecaster variables are chosen to split a node based on specific splitting conditions framed as optimization problems. Entropy is frequently employed as a common criterion for splitting in classification problems.

An example of applying Shannon's source coding theorem is the determination of the minimum length required for representing a random variable in bits. In a decision tree, entropy is calculated at each internal node as per **Eq 6**. Entropy is used to determine the best attribute of

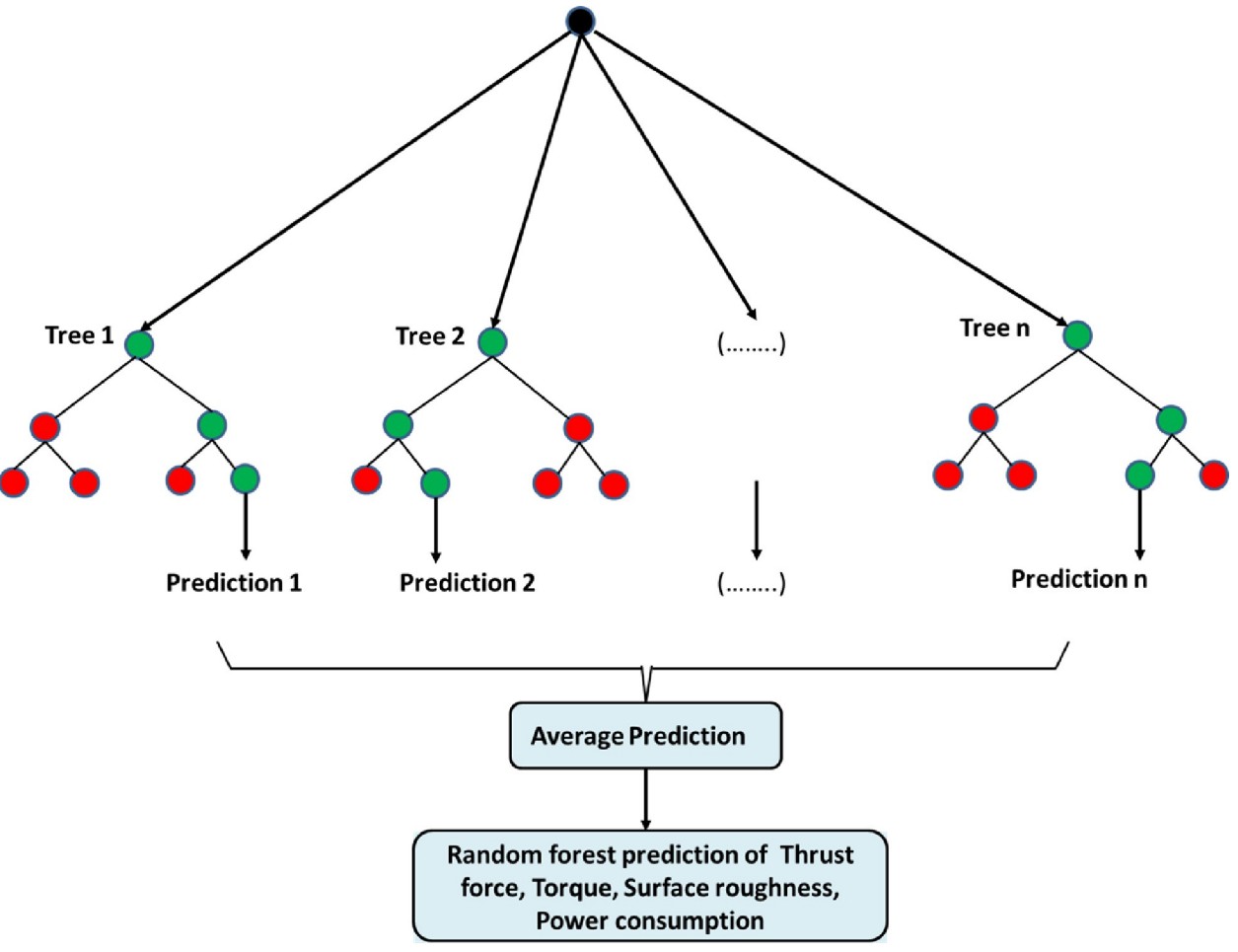

**Fig 5. Architecture of the random forest regression model used to predict power and surface roughness.**

the dataset to be used as a root of the decision tree. The attribute with minimum entropy is chosen as the root node, and the process is then repeated for each of the child nodes.

$$E = - \sum_{i=1}^{c} p_i \, X \, log(p_i) \tag{6}$$

Each class is assigned a prior probability, Pi, based on the total number of unique classes, C. For regression problems, the mean squared error is commonly utilized as the splitting criterion at each internal node to maximize information gain with each split in the decision tree. The optimal split is determined by minimizing the mean squared error and this process continues recursively until a stopping criterion like minimum samples in a leaf node or maximum tree depth is met. The outcome is a decision tree that can make predictions.

**2.4.4 SVR algorithm.** Fig 6 displays the arrangement of SVR, a data analysis technique that builds on statistical principles and expands SVM to tackle regression problems. It incorporates a hidden layer, automatically determined by the dataset, linking input and output layers in a manner reminiscent of an ANN model. As per Fig 7, SVR entails converting the feature vectors of sample data from a lower dimension to a higher dimension, followed by conducting analysis of regression by utilizing the kernel function in this high-dimensional space. Eq 7

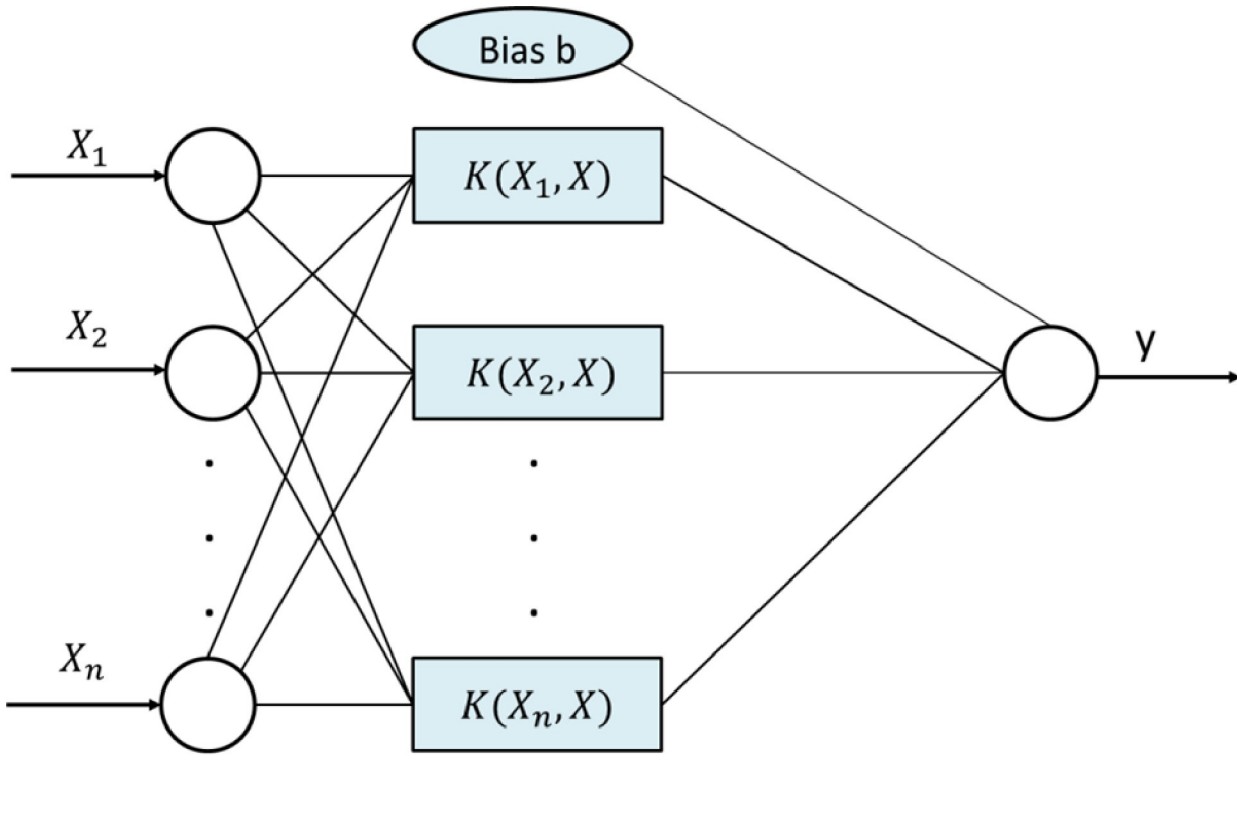

**Fig 6. Graphical representation of SVR.**

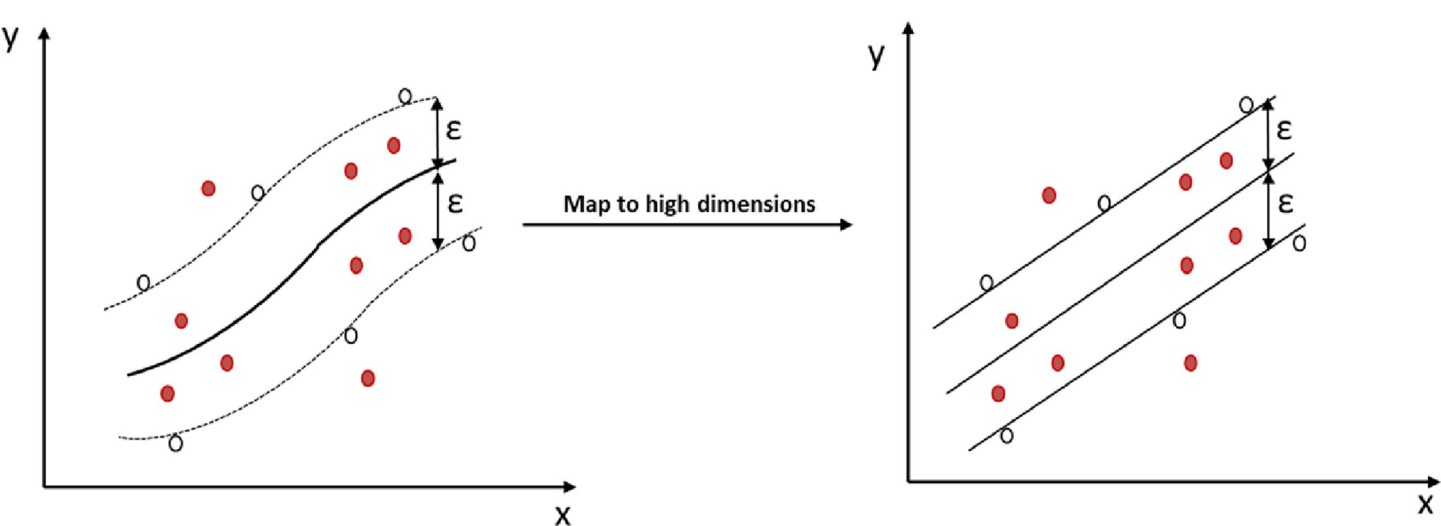

**Fig 7. Schematic digamma of the kernel trick.**

illustrates the governing principle of the support vector regression machine.

$$f(x) = \omega(x + b) \tag{7}$$

Where, $\omega$ represents the coefficient of friction, x denotes the input factor, and b stands for the bias constant. The last stage involves choosing the regression function that yields the greatest value. Once we have identified the best fit, we can utilize the model to make predictions on new data.

As the calculation proceeds, it will be necessary to determine which regression function has the highest value for each type of feature vector from the input as shown in **Eqs 8 and 9**.

$$\min \frac{1}{2}\omega^T\omega + C\frac{1}{N}\sum_{i=1}^{N} L(f(x_i), y_i) \tag{8}$$

$$L(y) = \begin{cases} 0, |f(x_i) - y_i| \leq \varepsilon \\ |f(x_i) - y_i| - \varepsilon, |f(x_i) - y_i| \geq \varepsilon \end{cases} \tag{9}$$

There are six variables in this model: C represents the penalty parameter; N is the number of samples; $f(x_i)$ indicates the predicted value of the feature vector for sample i; $y_i$ represents the actual value of the feature vector for sample i; L denotes the linear insensitive loss function, and $\varepsilon$ represents the maximum deviation. The dual form of SVR model can be derived by obtaining partial derivatives of these variables using both Lagrange equation and Karush-Kuhn-Tucker condition. The ultimate decision function is shown in Eq 10.

$$f(x) = \sum_{i=1}^{l} (\alpha_i^* - \alpha_i)K(x_i, x) + b \tag{10}$$

In this model, l denotes the number of Support vector regressor and $\alpha_i$ denotes the optimal solution. The kernel function K for nonlinear regression is expressed as $K(x_i, x) = \Phi(x_i) \times \Phi(x_j)$. To mitigate the problem of dimensional expansion in a high-dimensional space, this approach employs an enhanced kernel function. As a result, calculations are performed in a low-dimensional space before being mapped to a high-dimensional space.

## 2.5 Model performance evaluation

Training and testing sets were split using an 80:20 ratio, a widely accepted and established practice. To evaluate the effectiveness of the suggested model, three statistical measures were employed: Mean square error, mean absolute percentage deviation, and coefficient of determination ($R^2$). These indices are commonly used to gauge the accuracy of any data-driven modeling technique. R-squared ($R^2$) shows how well the model explains variance in the target, with higher values indicating a better fit. Mean Squared Error (MSE) highlights larger errors, while Mean Absolute Error (MAE) provides an average of absolute errors, less affected by outliers. Together, these metrics assess model performance in terms of fit, error size, and prediction accuracy. The $R^2$ value is computed using Eq 11.

$$R^2 = 1 - \frac{\sum_{i=1}^{n}(Y_{observed\ value} - Y_{predicted\ value})^2}{\sum_{i=1}^{n}(Y_{observed\ value} - \bar{Y})^2} \tag{11}$$

To assess the model effectiveness, the MSE is commonly employed, as demonstrated in Eq 12.

$$MSE = \frac{1}{n}\sum_{i=1}^{n}(y_P - y_A)^2 \tag{12}$$

The forecast values and the observed values are used to calculate the consistency of the neural network (NN) model result. The Mean Absolute Percentage Error (MAPE), determined by Eq 13, helps assess this.

$$MAPE = \frac{100\%}{n} \sum_{i=1}^{n} \left| \frac{y_\rho - y_A}{y_A} \right| \qquad (13)$$

## 3. Result and analysis

### 3.1 Experimental results

Table 5 displays the recorded response values and S/N ratios for Power and Surface roughness. The parameters of the study were first optimized based on the criteria of optimization. Power and Surface roughness represent the effort required to make a cut and surface quality of product respectively, so they were minimized in order to maximize performance. The optimized parameters were then used to evaluate the performance of the cutting process.

**Table 5. Investigational outcomes for power and surface roughness, along with the (S/N) ratios.**

| Run | Input Process Parameters | | | Power (KW) | | | | Surface Roughness (µm) | | | |
|---|---|---|---|---|---|---|---|---|---|---|---|
| | Additive Concentration (%) | Spindle Speed (RPM) | Feed Rate (mm/rev) | Replica 1 | Replica 2 | Mean Power (KW) | S/N ratios | Replica 1 | Replica 2 | Mean Surface Roughness (µm) | S/N ratios |
| 1 | 4 | 3053 | 0.1 | 0.21 | 0.29 | 0.25 | 11.99 | 1.28 | 1.24 | 1.26 | -2.01 |
| 2 | 4 | 3053 | 0.2 | 0.27 | 0.31 | 0.29 | 10.89 | 1.53 | 1.43 | 1.48 | -3.41 |
| 3 | 4 | 3053 | 0.3 | 0.34 | 0.28 | 0.31 | 10.09 | 1.79 | 1.69 | 1.74 | -4.81 |
| 4 | 4 | 4580 | 0.1 | 0.54 | 0.58 | 0.56 | 5.08 | 1.57 | 1.45 | 1.51 | -3.58 |
| 5 | 4 | 4580 | 0.2 | 0.68 | 0.62 | 0.65 | 3.69 | 1.67 | 1.59 | 1.63 | -4.24 |
| 6 | 4 | 4580 | 0.3 | 0.72 | 0.76 | 0.74 | 2.60 | 1.99 | 1.95 | 1.97 | -5.89 |
| 7 | 4 | 6107 | 0.1 | 0.85 | 0.89 | 0.87 | 1.26 | 1.58 | 1.46 | 1.52 | -3.64 |
| 8 | 4 | 6107 | 0.2 | 1.01 | 1.05 | 1.03 | -0.27 | 1.79 | 1.71 | 1.75 | -4.86 |
| 9 | 4 | 6107 | 0.3 | 1.18 | 1.24 | 1.21 | -1.65 | 2.28 | 2.22 | 2.25 | -7.04 |
| 10 | 6 | 3053 | 0.1 | 0.25 | 0.21 | 0.23 | 12.65 | 1.12 | 1.04 | 1.08 | -0.67 |
| 11 | 6 | 3053 | 0.2 | 0.25 | 0.27 | 0.26 | 11.57 | 1.24 | 1.22 | 1.23 | -1.80 |
| 12 | 6 | 3053 | 0.3 | 0.34 | 0.30 | 0.32 | 9.92 | 1.43 | 1.37 | 1.40 | -2.92 |
| 13 | 6 | 4580 | 0.1 | 0.44 | 0.36 | 0.40 | 8.05 | 1.49 | 1.35 | 1.42 | -3.05 |
| 14 | 6 | 4580 | 0.2 | 0.47 | 0.49 | 0.48 | 6.32 | 1.53 | 1.45 | 1.49 | -3.46 |
| 15 | 6 | 4580 | 0.3 | 0.65 | 0.61 | 0.63 | 4.00 | 1.71 | 1.65 | 1.68 | -4.51 |
| 16 | 6 | 6107 | 0.1 | 0.80 | 0.74 | 0.77 | 2.30 | 1.54 | 1.44 | 1.49 | -3.46 |
| 17 | 6 | 6107 | 0.2 | 0.92 | 0.90 | 0.91 | 0.83 | 1.76 | 1.72 | 1.74 | -4.81 |
| 18 | 6 | 6107 | 0.3 | 1.04 | 1.10 | 1.07 | -0.57 | 1.99 | 1.87 | 1.93 | -5.71 |
| 19 | 8 | 3053 | 0.1 | 0.24 | 0.18 | 0.21 | 13.61 | 1.09 | 1.01 | 1.05 | -0.42 |
| 20 | 8 | 3053 | 0.2 | 0.25 | 0.21 | 0.23 | 12.65 | 1.32 | 1.26 | 1.29 | -2.21 |
| 21 | 8 | 3053 | 0.3 | 0.38 | 0.34 | 0.36 | 8.82 | 1.50 | 1.40 | 1.45 | -3.23 |
| 22 | 8 | 4580 | 0.1 | 0.41 | 0.43 | 0.42 | 7.46 | 1.35 | 1.31 | 1.33 | -2.48 |
| 23 | 8 | 4580 | 0.2 | 0.52 | 0.50 | 0.51 | 5.91 | 1.51 | 1.43 | 1.47 | -3.35 |
| 24 | 8 | 4580 | 0.3 | 0.58 | 0.52 | 0.55 | 5.16 | 1.68 | 1.56 | 1.62 | -4.19 |
| 25 | 8 | 6107 | 0.1 | 0.76 | 0.84 | 0.80 | 1.96 | 1.45 | 1.31 | 1.38 | -2.80 |
| 26 | 8 | 6107 | 0.2 | 0.90 | 0.84 | 0.87 | 1.26 | 1.73 | 1.63 | 1.68 | -4.51 |
| 27 | 8 | 6107 | 0.3 | 0.98 | 1.04 | 1.01 | -0.11 | 1.96 | 1.92 | 1.94 | -5.76 |

Among the parameters examined in Table 6, it was determined that spindle speed had the most significant impact on Power, with feed rate and additive concentration following closely behind. Despite a lesser effect than speed and feed rate, additive concentration also played a role.

Figs 8 and 9 illustrates the main effects plot for the S/N ratios for Power and Surface roughness respectively. This indicates that the magnitude of power keeps increasing as the speed of cutting is increased from 3053 to 6107 rpm. This suggests that the speed of cutting influences more strain on the tool and work-piece contact zone, increasing the magnitude of power, implying a higher level of strain and power. Moreover, power decreased when the additive concentration increased from 6% to 8%. This demonstrates that the combination of GS440L and TMPTO can significantly reduce the amount of tool strain and power when cutting at higher speeds due their synergetic effect. As a result, the addition of GS440L and TMPTO to the tool can improve the cutting quality. In addition, the addition of these materials also helps reduce the cutting zone temperature, thus extending tool life. Furthermore, the improved cutting properties help increase productivity.

It has been shown that the texture of surfaces is a critical factor in assessing the machining performance of a work-piece. Higher spindle speed and feed rates led to an increase in the Ra value. There is a suggestion that rough surface texture at high speeds may be due to intense vibrations. Meanwhile, the higher chip thickness associated with increased feed rates could be responsible for the rise in Ra value. As indicated in Table 6, feed rate has the greatest noteworthy effect on surface roughness, followed by speed and additive concentration.

Fig 9 shows the main effects plot for the S/N ratios of surface roughness, indicating that the level of surface roughness varies with different variables. Another factor influencing surface roughness was the feed rate. A lower feed rate of 0.1 mm/rev led to minimal tool advancement relative to the workpiece, resulting in very fine feed marks and a smoother appearance on the surface. On increasing it to 0.3 mm/rev, fewer but stronger feed marks appeared within each unit area, leading to surface roughness increases due to greater particle impact from faster cutting tool advancement causing fracturing on the surface.

When it comes to Power Analysis, the data in Table 7 reveals that spindle speed has the greatest impact on Power (87.89%), followed feed rate (6.96%) and additive concentration (2.98%). This suggests the most significant factor is cutting speed that affecting power. When the feed rate is increased and the concentration of additives is increased, the power output will be less affected. In spite of this, a blend of TMPTO and GS440L reduces the power produced during high speed drilling operation when the additive concentration is boosted.

**Table 6. The S/N ratio table for power and surface roughness.**

|  | Level | Additive | Spindle | Feed Rate |
|---|---|---|---|---|
|  |  | Concentration (%) | Speed (RPM) | (rev/mm) |
|  | 1 | 4.8541 | 11.3543 | 7.1508 |
|  | 2 | 6.1193 | 5.3643 | 5.8726 |
| Power | 3 | 6.3021 | 0.5569 | 4.252 |
|  | Delta | 1.448 | 10.7974 | 2.8988 |
|  | Rank | 3 | 1 | 2 |
| Surface Roughness | 1 | -4.386 | -2.386 | -2.456 |
|  | 2 | -3.377 | -3.860 | -3.627 |
|  | 3 | -3.215 | -4.732 | -4.895 |
|  | Delta | 1.171 | 2.346 | 2.440 |
|  | Rank | 3 | 2 | 1 |

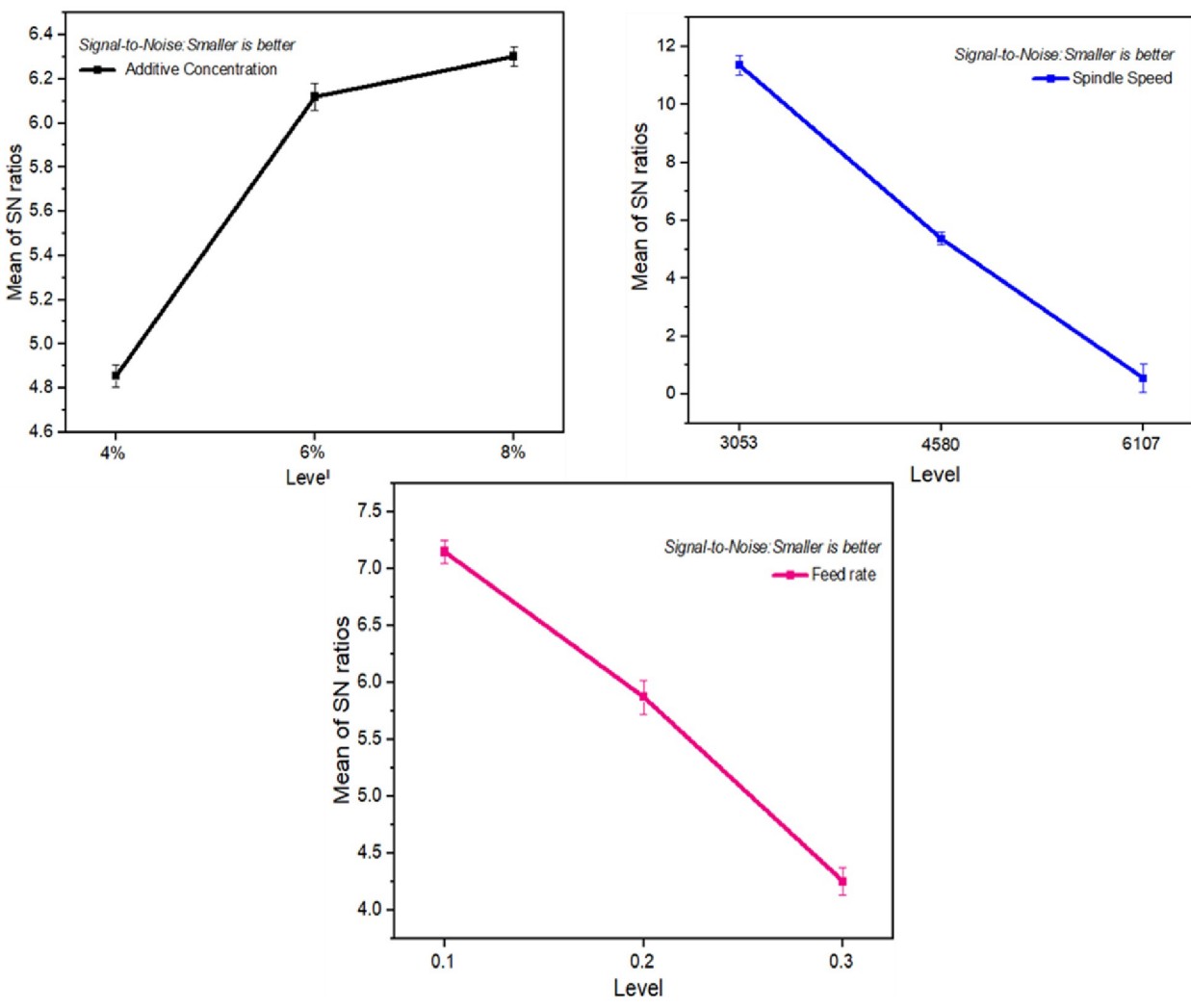

**Fig 8. Main effects plot for S/N ratios of power.**

In the case of analyzing surface roughness, Table 7 shows that the feed rate has the most significant influence on surface roughness (43.51%), followed by speed (38.48%) and additive concentration (11.90%). This suggests that both feed rate and spindle speed are critical factors affecting surface roughness. Nevertheless, using a combination of TMPTO and GS440L reduces the surface roughness during high-speed drilling when increasing the additive concentration, indicating its substantial impact on surface roughness. Thus, achieving optimal surface roughness depends on finding an optimal combination of feed rate, spindle speed, and additive concentration.

### 3.2 Forecasting of outcomes using various ML models

In machine learning, the training-to-testing data ratio varies with context and data size. A common ratio is 80:20, but for larger datasets, 90:10 or 95:5 may be used. The goal is to balance having enough data to train the model while reserving enough for effective testing to evaluate its generalization. Four distinct ML (machine learning) models based on regression were utilized to forecast response factors for high-speed drilling processes. A dataset of fifty four data

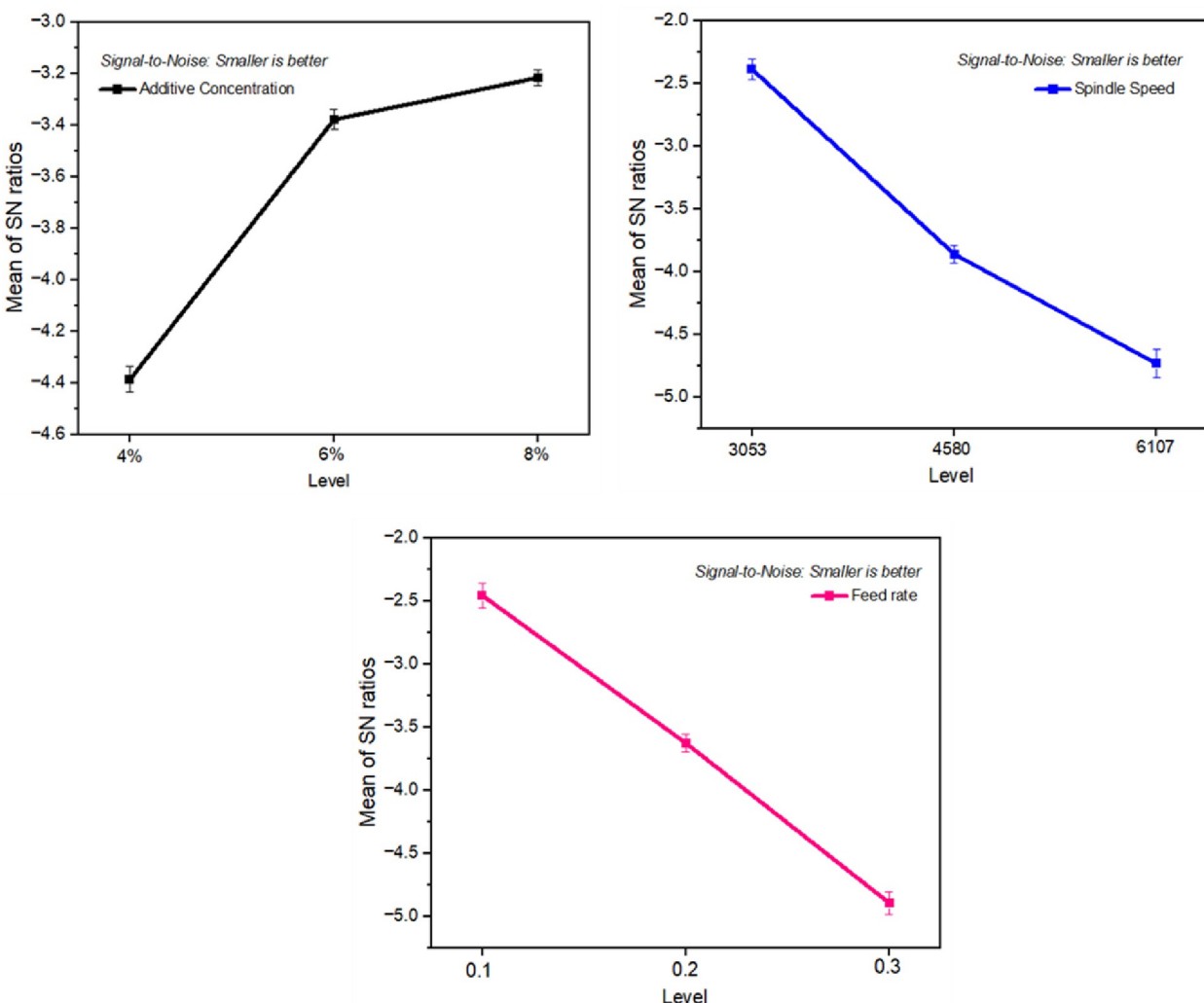

**Fig 9. Main effects plot for S/N ratios of surface roughness.**

points obtained from investigational data was used in developing and evaluating the model, employing regression-based machine learning techniques. Three different predictors, namely concentration of additive, feed rate and spindle speed are utilized to forecast responses using

**Table 7. ANOVA table for power and surface roughness.**

|  | Source | DF | Adj SS | Adj MS | F-Value | P-Value | % Contribution |
|---|---|---|---|---|---|---|---|
|  | Concentration (%) | 2 | 0.06983 | 0.02975 | 9.76 | 0.001 | 2.98 |
|  | Spindle Speed (RPM) | 2 | 2.05973 | 1.02986 | 337.66 | 0.000 | 87.89 |
| Power | Feed Rate (rev/mm) | 2 | 0.16315 | 0.08157 | 26.75 | 0.000 | 6.96 |
|  | Error | 20 | 0.051 | 0.00305 |  |  | 2.17 |
|  | Total | 26 | 2.34338 |  |  |  |  |
| Surface Roughness | Concentration (%) | 2 | 0.2369 | 0.118426 | 19.41 | 0.000 | 11.88 |
|  | Spindle Speed (RPM) | 2 | 0.7668 | 0.383393 | 62.84 | 0.000 | 38.48 |
|  | Feed Rate (rev/mm) | 2 | 0.8671 | 0.433526 | 71.05 | 0.000 | 43.51 |
|  | Error | 20 | 0.1220 | 0.006101 |  |  | 6.10 |
|  | Total | 26 | 1.9927 |  |  |  |  |

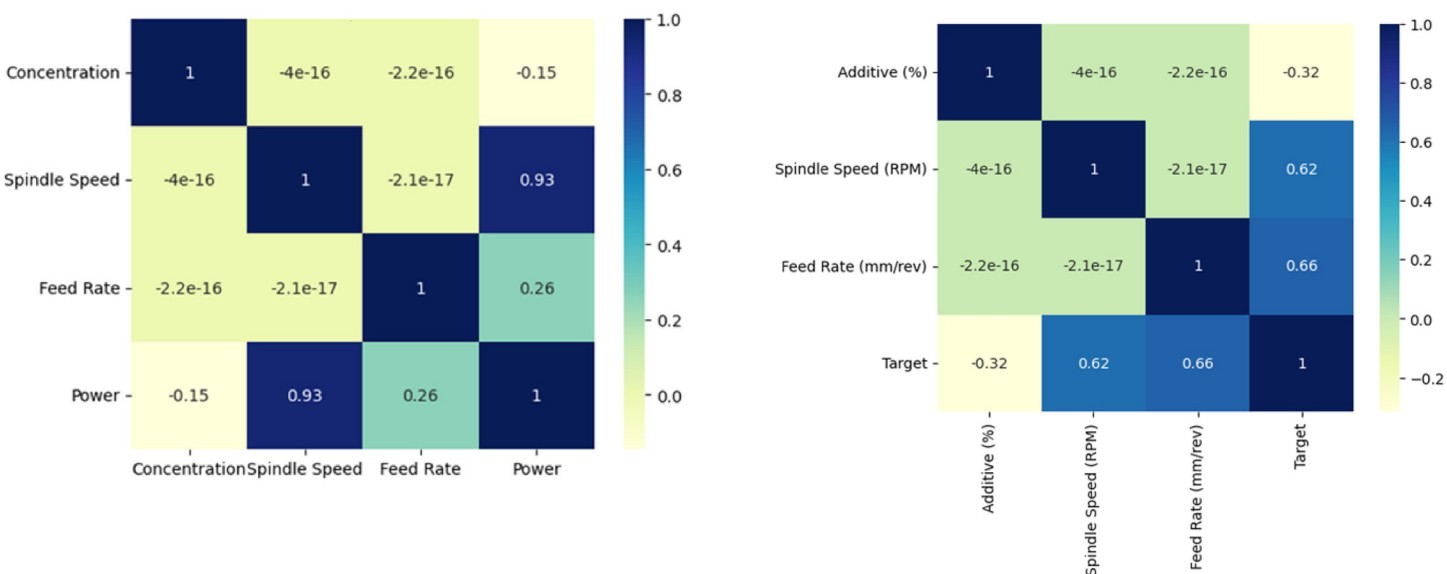

**Fig 10.** **(a)** Heat map of the Pearson correlation coefficient matrix for Power. **b)** Heat map of the Pearson correlation coefficient matrix for Surface Roughness.

various models. To standardize the data and mitigate errors stemming from input parameter variations in units, a standard scaling function is applied to both training and testing datasets. The model undergoes cross-validation with GridSearch CV to ensure optimal parameters and avoid overfitting or underfitting. By assessing training and test errors, the model's fit is determined. The input data was divided into two parts: training and testing. Training involved randomly selecting 80% of the input data, while testing involved randomly selecting 20%. The given data points were used to evaluate the model's accuracy and error rate.

**3.2.1 Data preprocessing and model performance.** Based on the Pearson correlation coefficients, the correlations between the factors influencing The Power (a), and surface roughness (b) are shown in Fig 12(A), and 12(B) respectively. Additionally, the results suggest that there is no strong correlation between any independent variable.

It can be seen from the **Fig 10(A),** the spindle speed appears to be most closely related to power. **Fig 10(B)** shows spindle speed appears to be most closely related to surface roughness followed by feed arte. It is possible to determine the extent to which each of the influencing factors contributes to power and surface roughness using the Pearson correlation coefficient, but it cannot be used to determine the degree to which each factor contributes to power and surface roughness. To identify the key controlling features affecting power and surface roughness by analyzing their respective weights, a machine learning method was also applied to determine the degree of nonlinear influence on the power and surface roughness.

**Table 8. Performance metrics for power.**

| Models/Performance Metrics | Test data result | | | Training data result | | |
|---|---|---|---|---|---|---|
| | **R-Squared** | **MSE** | **MAE** | **R-Squared** | **MSE** | **MAE** |
| Random Forest | 0.80 | 0.0035 | 0.0438 | 0.98 | 0.0015 | 0.0295 |
| Linear Regression | 0.93 | 0.0056 | 0.0650 | 0.96 | 0.0030 | 0.0424 |
| SVM | 0.47 | 0.1077 | 0.2506 | 0.97 | 0.0243 | 0.1240 |
| Decision Tree | 0.95 | 0.0040 | 0.0483 | 1.00 | 0.0000 | 0.0000 |

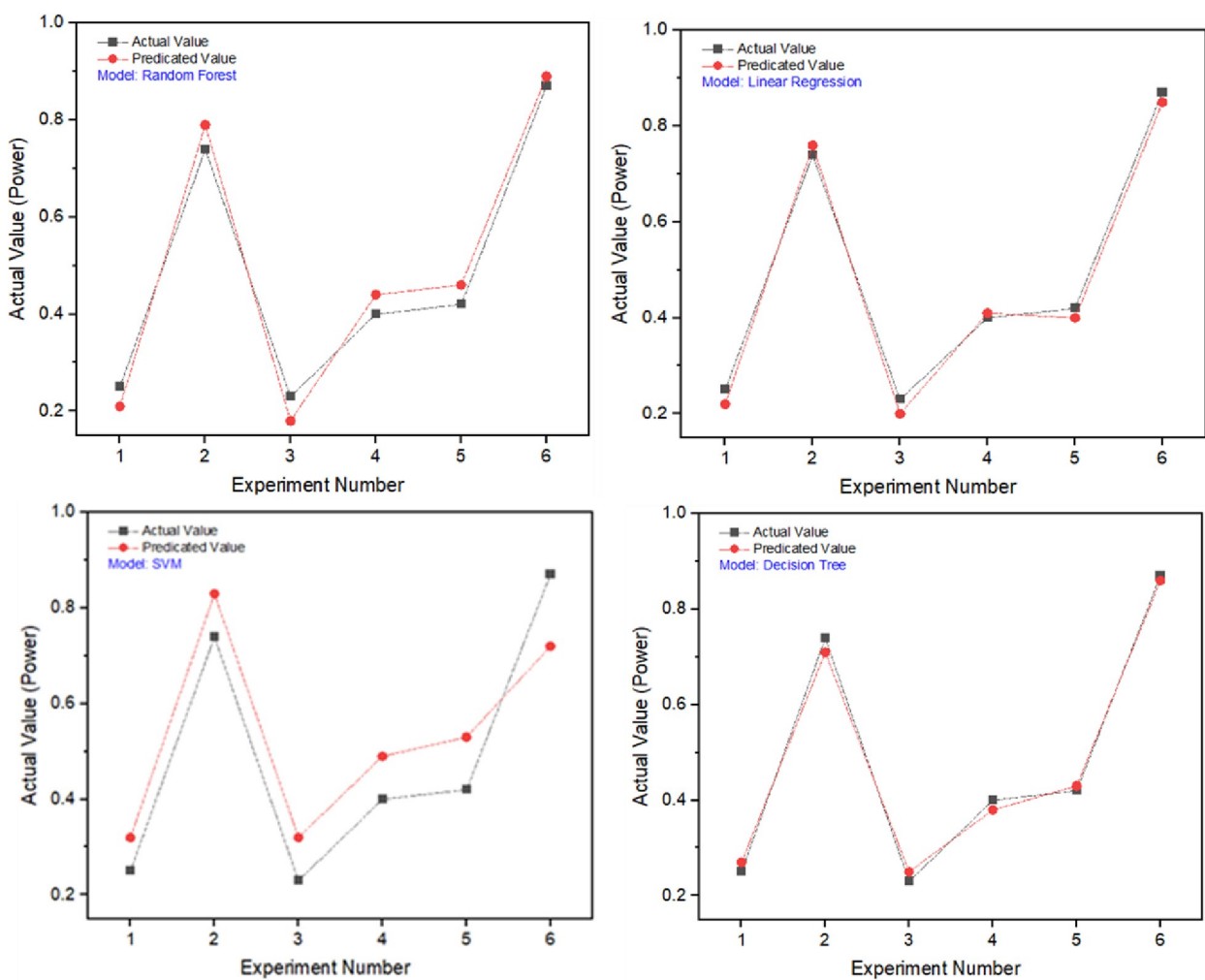

**Fig 11. Model predicted and actual power values.**

In **Table 8**, the four machine learning models are compared based on their performance. The experiment shows that Decision Tree (95%) and linear regression (93%) are the most appropriate models for Power prediction. The outcomes also indicate that the model predictions are very nearby to the experimental values. The outcomes revealed that the Decision Tree had the greatest performance with a mean absolute error of 0.0483. In contrast, linear regression had a mean absolute error of 0.0650. The other two models had mean absolute errors of 0.0438 and 0.2506 respectively.

**Table 9. Performance metrics for Surface roughness.**

| Models/Performance Metrics | Test data result | | | Training data result | | |
|---|---|---|---|---|---|---|
| | R-Squared | MSE | MAE | R-Squared | MSE | MAE |
| Random Forest | 0.85 | 0.0088 | 0.0786 | 0.93 | 0.0054 | 0.0543 |
| Linear Regression | 0.96 | 0.0034 | 0.0514 | 0.98 | 0.0072 | 0.0736 |
| SVM | 0.86 | 0.2478 | 0.2850 | 0.93 | 0.2020 | 0.1768 |
| Decision Tree | 0.68 | 0.0065 | 0.0683 | 1.00 | 0.0000 | 0.0000 |

Fig 11 compares the actual power against the predicted power. This comparison shows that the four models provide accurate predictions of power. Decision Tree and Linear regression model appear to be the best precise models for predicting power based on the results. In addition to the high correlation coefficients, which indicate that there is a strong linear relationship between the predicted power and the actual power, this is also supported by the results. It is also noteworthy to point out that Random forest model also possesses a good correlation coefficient, which indicates that it may be a viable means of predicting power.

Table 9 displays the discrepancies related to testing and training for surface roughness, including three performance measures utilized to assess the precision of the projected models. The linear regression model attained a 96% accuracy rate on the test data and 98% on the training data.

This demonstrated the model's effectiveness in predicting surface roughness. In addition, it showed the MSE (mean square error) of 0.0034 and MAE (mean absolute error) of 0.0514 showing the efficiency of the proposed approach. As well as this, another models such as Random forest and SVM also exhibits an accurate prediction of the surface roughness with a higher than 80% accuracy rate.

In Fig 12, the actual surface roughness is contrasted with the anticipated surface roughness value. The comparison shows that the four models provide accurate predictions of the surface roughness. The model with the highest correlation coefficient is Model 2 (Linear Regression), with a correlation coefficient of 0.96. This indicates that Model 2 is the most accurate model for predicting surface roughness. High speed metal cutting can benefit greatly from the use of this method, as it increases the efficiency of the machining process.

This investigation evaluated the feasibility of vegetable oil-based blends in general machining conditions. For a more comprehensive assessment, factors like tool geometry, workpiece material properties, and environmental conditions should be considered. This would provide deeper insights into their performance in real-world scenarios and aid in refining their industrial applications.

## 4. Conclusion

This study aimed to investigate the power consumption and surface roughness of Al 6061 materials during high-speed drilling, employing TMPTO lube oil with chemically modified sulfurized vegetable oil olefins at concentrations of 4%, 6%, and 8%. The following conclusions can be drawn:

The range of additive concentrations demonstrates a synergistic effect with TMPTO, contributing to reduced energy consumption and improved surface finish.

ANOVA analysis indicates that spindle speed has the most significant impact on Power (87.89%), followed by feed rate (6.96%) and additive concentration (2.98%). In contrast, feed rate (43.51%) exerts the most notable influence on surface roughness, followed by speed (38.48%) and additive concentration (11.90%). Notably, adjusting additive concentration has a more pronounced effect on surface quality than on power consumption. Experimental evidence and machine learning predictions confirm significant correlations, with the proposed model achieving satisfactory outcomes and a high level of prediction accuracy.

Four machine learning models were compared based on their performance. Decision Tree (95%) and Linear Regression (93%) were found to be the most suitable models for predicting Power values. For surface roughness prediction, Linear Regression (96%), Support Vector Machine (86%), and Random Forest (85%) exhibited the most appropriate performance.

Predictive models have proven highly beneficial in selecting suitable parameters for machining applications. This approach is expected to be a valuable tool for researchers and

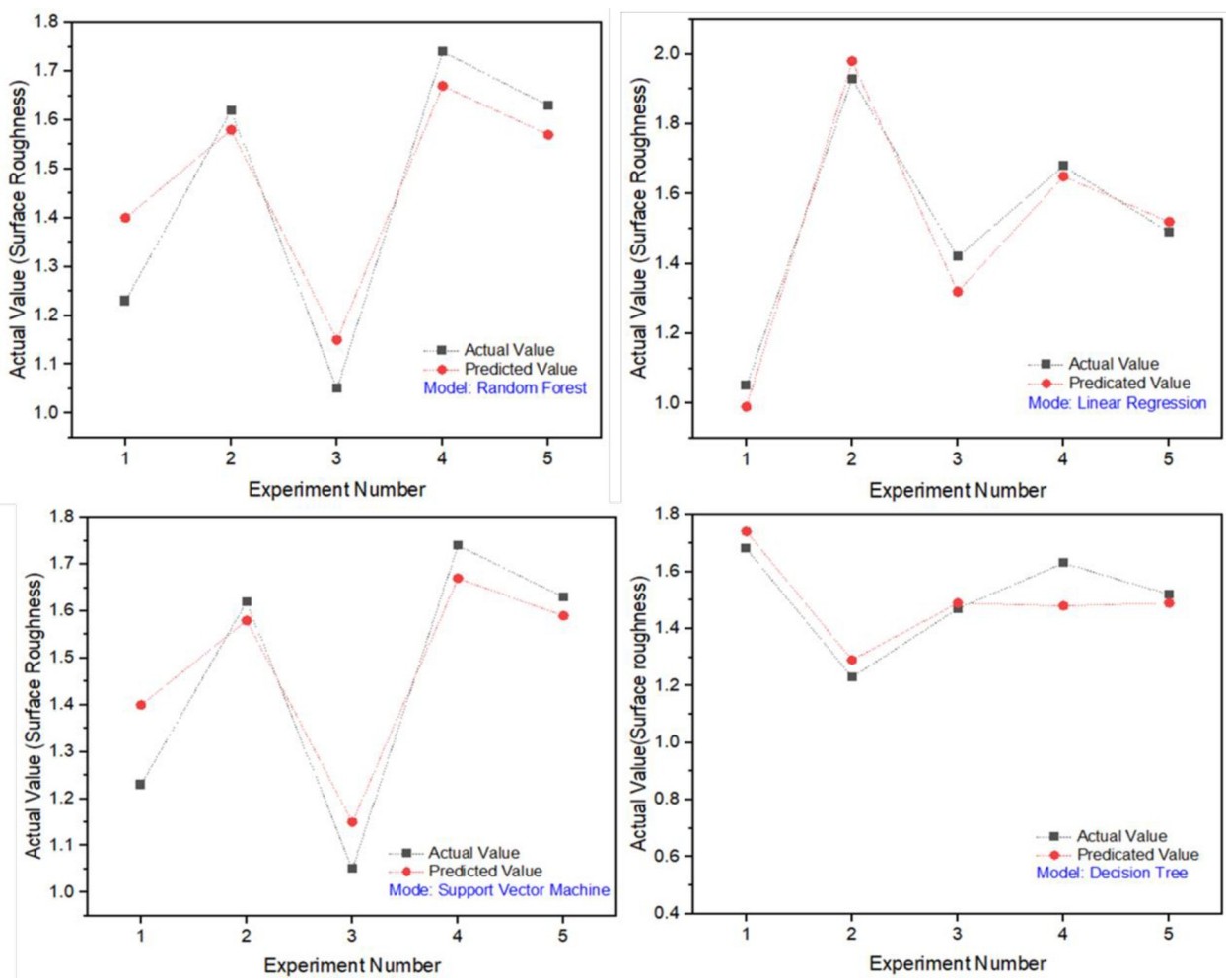

**Fig 12. Model predicted and actual surface roughness values.**

industry alike. By leveraging predictive models, researchers and industries can efficiently and accurately identify optimal process parameters pertaining to the specific lube oil with an objective to have better surface quality with reduced power.

## Author Contributions

**Conceptualization:** Pramod S. Kathmore, Ondřej Mizera.

**Data curation:** Pramod S. Kathmore, Duran Kaya, Lenka Cepova, Ondřej Mizera.

**Formal analysis:** Pramod S. Kathmore, Duran Kaya, Lenka Cepova, Ondřej Mizera.

**Funding acquisition:** Pramod S. Kathmore, Lenka Cepova, Ondřej Mizera.

**Investigation:** Pramod S. Kathmore, Bhanudas D. Bachchhav, Sachin Salunkhe.

**Methodology:** Pramod S. Kathmore, Bhanudas D. Bachchhav, Sachin Salunkhe.

**Project administration:** Bhanudas D. Bachchhav, Duran Kaya, Ondřej Mizera, Emad Abouel Nasr.

**Resources:** Sachin Salunkhe, Lenka Cepova, Ondřej Mizera, Emad Abouel Nasr.

**Software:** Sachin Salunkhe, Lenka Cepova, Ondřej Mizera, Emad Abouel Nasr.

**Supervision:** Bhanudas D. Bachchhav, Sachin Salunkhe, Lenka Cepova, Emad Abouel Nasr.

**Validation:** Duran Kaya, Sachin Salunkhe, Lenka Cepova, Emad Abouel Nasr.

**Visualization:** Bhanudas D. Bachchhav, Duran Kaya, Sachin Salunkhe, Lenka Cepova.

**Writing – original draft:** Bhanudas D. Bachchhav, Duran Kaya.

**Writing – review & editing:** Bhanudas D. Bachchhav, Duran Kaya.

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
