## [Decision Letter · Decision Letter 0]

6 Sep 2024

PONE-D-24-34684Analyzing the Efficacy of Trimethylolpropane Trioleate Oil for Predicting Cutting Power and Surface Roughness in High-Speed Drilling of Al-6061 through Machine LearningPLOS ONE

Dear Dr. Salunkhe,

Thank you for submitting your manuscript to PLOS ONE. After careful consideration, we feel that it has merit but does not fully meet PLOS ONE’s publication criteria as it currently stands. Therefore, we invite you to submit a revised version of the manuscript that addresses the points raised during the review process.

We look forward to receiving your revised manuscript.

Kind regards,

Himadri Majumder, Ph.D

Academic Editor

PLOS ONE

Journal Requirements:

   "The authors would like to express their gratitude to King Saud University, Riyadh, Saudi Arabia, for sponsoring this work under Researchers Supporting Project number (RSP2024R164). This article was co-funded by the European Union under the REFRESH – Research Excellence For Region Sustainability and High-tech Industries project number CZ.10.03.01/00/22_003/0000048 via the Operational Programme Just Transition and has been done in connection with project Students Grant Competition SP2024/087 „Specific Research of Sustainable Manufacturing Technologies“ financed by the Ministry of Education, Youth and Sports and Faculty of Mechanical Engineering VŠB-TUO. Article has been done in connection with project Students Grant Competition SP2024/087 „Specific Research of Sustainable Manufacturing Technologies“ financed by the Ministry of Education, Youth and Sports and Faculty of Mechanical Engineering VŠB-TUO."

6. We note that your Data Availability Statement is currently as follows: All relevant data are within the manuscript and its Supporting Information files.

Additional Editor Comments:

Reviewers' comments on your paper entitled "Analyzing the Efficacy of Trimethylolpropane Trioleate Oil for Predicting Cutting Power and Surface Roughness in High-Speed Drilling of Al-6061 through Machine Learning" have now been received. Based on the reviewers' recommendations, the manuscript requires MINOR REVISION. Please pay attention to all the comments and make all necessary changes. In order to expedite the processing of the revised manuscript, please be as specific as possible in your response to the reviewers. Please include with your revised submission an itemized, point-by-point response to the reviewers which details the changes made.

Reviewers' comments:

Reviewer's Responses to Questions

**Comments to the Author**

1. Is the manuscript technically sound, and do the data support the conclusions?

Reviewer #1: Yes

Reviewer #2: Yes

2. Has the statistical analysis been performed appropriately and rigorously? 

Reviewer #1: Yes

Reviewer #2: Yes

3. Have the authors made all data underlying the findings in their manuscript fully available?

Reviewer #1: Yes

Reviewer #2: Yes

4. Is the manuscript presented in an intelligible fashion and written in standard English?

Reviewer #1: Yes

Reviewer #2: Yes

5. Review Comments to the Author

Reviewer #1: I wish to congratulate the authors for this novel attempt and construction of manuscript in best manner.

I wish to raise the following comments for further understanding.

1) What are the reasons for selecting TMPTO as base oil for machining?

2) What is the additive concentration in the base oil and on what basis are these selected?

3) Can you tell me how you performed the anti-wear and extreme pressure tests?

4) What should be the ratio of training data to testing data in machine learning?

5) What were the reasons for selecting aluminum material for investigating samples under drilling operation?

6) What assumptions did you consider when selecting machine learning models?

7) What is the significance of R-squared, MSE and MAE in machine learning?

Reviewer #2: This study offers valuable insights into the use of TMPTO as a cutting fluid, with a focus on parameters like additive concentration, spindle speed, and feed rate. However, several limitations reduce the overall impact and applicability of the findings.

1.The study primarily examines specific parameters while overlooking critical factors such as tool geometry, material properties of the workpiece, and environmental conditions. This limited scope may restrict the generalizability of the results to broader industrial applications.

2. The study employs machine learning techniques to predict cutting performance, but the reliability of these models depends heavily on the quality and quantity of the data used for training. There is a risk of overfitting or underfitting due to limited or biased datasets, and the study does not address the potential inaccuracies that could arise from unaccounted variables in the drilling process.

3. A significant drawback is the absence of a comprehensive comparative analysis with other established cutting fluids, particularly mineral-based alternatives. This omission makes it difficult to fully evaluate the effectiveness of TMPTO as a sustainable option.

4. While the study highlights the environmental benefits of TMPTO, it does not explore the economic implications of transitioning from traditional cutting fluids. Key factors such as cost-effectiveness, availability, and long-term sustainability are not addressed, which could impact the practical adoption of TMPTO in industrial settings.

5. The study focuses on power consumption and surface roughness but does not adequately consider the long-term effects of TMPTO on tool wear and lifespan. Understanding these effects is crucial for determining the viability of TMPTO as a cutting fluid.

6. Despite advocating for environmentally friendly lubricants, the study does not discuss potential safety and health risks associated with TMPTO use in industrial environments. This oversight should be addressed to ensure the comprehensive evaluation of TMPTO.

6. PLOS authors have the option to publish the peer review history of their article (what does this mean?). If published, this will include your full peer review and any attached files.

Reviewer #1: No

Reviewer #2: **Yes: **vishal Naranje

---

## [Author Response · Author response to Decision Letter 0]

26 Sep 2024

Sept. 9, 2024

Dear Dr. Himadri Majumder, 

Academic Editor

PLOS ONE

Enclosed is a carefully modified Manuscript ID: PONE-D-24-34684, entitled “Analyzing the Efficacy of Trimethylolpropane Trioleate Oil for Predicting Cutting Power and Surface Roughness in High-Speed Drilling of Al-6061 through Machine Learning”. 

The authors are thankful to the reviewers for making constructive suggestions to improve the quality of the paper. The paper is thoroughly revised and the required changes are highlighted by coloured text in the revised manuscript. Here we present reply to reviewer comments.

Reviewer #1:

Comment 1: What are the reasons for selecting TMPTO as base oil for machining? 

Reply: Thank you for pointing out technical briefings. Following changes are made in revised manuscript. 

TMPTO was chosen as the base oil for machining due to its outstanding performance and eco-friendliness. It provides stable viscosity across temperatures, excellent low-temperature fluidity, and a high flash point for safety. With strong hydrolytic and foaming stability, it ensures prolonged use while enhancing tool performance and reducing wear. As a biodegradable alternative to mineral oils, TMPTO is a sustainable choice for machining.

Comment 2: What is the additive concentration in the base oil and on what basis are these selected? 

Reply: Thank you for pointing out technical briefings. Most additive suppliers recommend a 2%–4% concentration for optimal performance. To validate this, we tested various concentrations, focusing on antiwear performance. These tests helped identify the ideal range for maximum protection and efficiency in machining operations.

Comment 3: Can you tell me how you performed the anti-wear and extreme pressure tests? 

Reply: We appreciate you bringing this inconsistency to light. The results of AW and EP tests are depicted in revised manuscript. An anti-wear assessment was conducted in accordance with the ASTM D4172 standard, while an extreme pressure evaluation was performed following the ASTM D2783 standard.

Comment 4: What should be the ratio of training data to testing data in machine learning? 

Reply: Thank you for pointing out technical briefings. We have revised the manuscript as per reviewer’s suggestion. 

In machine learning, the training-to-testing data ratio varies with context and data size. A common ratio is 80:20, but for larger datasets, 90:10 or 95:5 may be used. The goal is to balance having enough data to train the model while reserving enough for effective testing to evaluate its generalization.

Comment 5: What were the reasons for selecting aluminum material for investigating samples under drilling operation? 

Reply: Thank you for pointing out technical briefings. Reasons for selecting aluminum material for investigating samples under drilling operation are mentioned in revised manuscript.

In metal cutting, cooling is often prioritized over lubrication. However, in high-speed cutting of ductile, non-ferrous metals like aluminum alloys, lubrication becomes essential. It reduces friction, aids cooling, and prevents chip buildup, which can impair cutting performance, accuracy, and surface finish. Cutting fluids are crucial for both cooling and lubricating, ensuring optimal conditions and tool performance by removing debris effectively.

Comment 6: What assumptions did you consider when selecting machine learning models? 

Reply: Thank you for pointing out technical briefings. Machine learning models rely on several key assumptions, including: independent observations, identically distributed data, linearity (for linear models), relevant features, no multicollinearity, normally distributed errors (in some models), model simplicity, sufficient data, and handling noisy data. These assumptions are considered for proper model performance and interpretation.

Comment 7: What is the significance of R-squared, MSE and MAE in machine learning? 

Reply: Thank you for pointing out technical briefings. R-squared (R²) shows how well the model explains variance in the target, with higher values indicating a better fit. Mean Squared Error (MSE) highlights larger errors, while Mean Absolute Error (MAE) provides an average of absolute errors, less affected by outliers. Together, these metrics assess model performance in terms of fit, error size, and prediction accuracy.

Reviewer #2:

Comment 1: The study primarily examines specific parameters while overlooking critical factors such as tool geometry, material properties of the workpiece, and environmental conditions. This limited scope may restrict the generalizability of the results to broader industrial applications. 

Reply: We appreciate you bringing this inconsistency to light. 

This investigation evaluated the feasibility of vegetable oil-based blends in general machining conditions. For a more comprehensive assessment, factors like tool geometry, workpiece material properties, and environmental conditions should be considered. This would provide deeper insights into their performance in real-world scenarios and aid in refining their industrial applications. This is considered as a future scope of work.

Comment 2: The study employs machine learning techniques to predict cutting performance, but the reliability of these models depends heavily on the quality and quantity of the data used for training. There is a risk of overfitting or underfitting due to limited or biased datasets, and the study does not address the potential inaccuracies that could arise from unaccounted variables in the drilling process. 

Reply: We appreciate you bringing this inconsistency to light. It’s true that machine learning models rely heavily on the quantity and quality of input data, we can still optimize performance with limited datasets by applying various preprocessing techniques before feeding the data into the model. In this investigation, the data was cleaned and preprocessed using techniques such as normalization, scaling, handling missing data, feature selection, cross-validation, regularization to prevent overfitting by penalizing large coefficients, and ensemble methods.

Comment 3: A significant drawback is the absence of a comprehensive comparative analysis with other established cutting fluids, particularly mineral-based alternatives. This omission makes it difficult to fully evaluate the effectiveness of TMPTO as a sustainable option. 

Reply: Thank you for pointing out technical briefings. You’ve raised a valid point about the lack of a comparative analysis with mineral-based cutting fluids. However, the focus of this research is to assess the lubricant efficacy in high-speed drilling of Al-6061 for energy savings and improved surface quality. A comparative analysis is beyond the scope of this manuscript and has been omitted due to length constraints. Future research could explore the effectiveness of TMPTO as a sustainable alternative. 

Comment 4: While the study highlights the environmental benefits of TMPTO, it does not explore the economic implications of transitioning from traditional cutting fluids. Key factors such as cost-effectiveness, availability, and long-term sustainability are not addressed, which could impact the practical adoption of TMPTO in industrial settings. 

Reply: Thank you for your valuable feedback. We agree that exploring the economic implications of transitioning from traditional cutting fluids to TMPTO is essential for a comprehensive evaluation. Factors such as cost-effectiveness, availability, and long-term sustainability are indeed critical for its practical adoption. In future research, we plan to incorporate a detailed economic analysis to address these aspects and provide a more holistic view of TMPTO's feasibility in industrial settings.

Comment 5: The study focuses on power consumption and surface roughness but does not adequately consider the long-term effects of TMPTO on tool wear and lifespan. Understanding these effects is crucial for determining the viability of TMPTO as a cutting fluid. 

Reply: Thank you for your thoughtful feedback. You are absolutely right that the long-term effects of TMPTO on tool wear and lifespan are crucial factors in assessing its overall viability as a cutting fluid. While this study primarily focused on power consumption and surface roughness, we recognize the importance of tool wear analysis and plan to include this aspect in future studies to provide a more comprehensive evaluation of TMPTO’s performance. However, preliminary tribological evaluation of Anti-wear and Extreme Pressure characteristics of formulated samples are incorporated in revised manuscript. 

Comment 6: Despite advocating for environmentally friendly lubricants, the study does not discuss potential safety and health risks associated with TMPTO use in industrial environments. This oversight should be addressed to ensure the comprehensive evaluation of TMPTO. 

Reply: Thank you for your critical observation. This study emphasized environmental and performance aspects, while recognizing that safety is crucial. TMPTO, derived from vegetable oils, was chosen to develop environmentally friendly lubricants with reduced health hazards.

---

## [Decision Letter · Decision Letter 1]

9 Oct 2024

Analyzing the Efficacy of Trimethylolpropane Trioleate Oil for Predicting Cutting Power and Surface Roughness in High-Speed Drilling of Al-6061 through Machine Learning

PONE-D-24-34684R1

Dear Dr. Salunkhe,

We’re pleased to inform you that your manuscript has been judged scientifically suitable for publication and will be formally accepted for publication once it meets all outstanding technical requirements.

Kind regards,

Himadri Majumder, Ph.D

Academic Editor

PLOS ONE

Additional Editor Comments (optional):

As per the reviewers' feedback, I am happy to recommend "Acceptance" of this revised paper.

Reviewers' comments:

Reviewer's Responses to Questions

**Comments to the Author**

1. If the authors have adequately addressed your comments raised in a previous round of review and you feel that this manuscript is now acceptable for publication, you may indicate that here to bypass the “Comments to the Author” section, enter your conflict of interest statement in the “Confidential to Editor” section, and submit your "Accept" recommendation.

Reviewer #1: All comments have been addressed

Reviewer #2: All comments have been addressed

2. Is the manuscript technically sound, and do the data support the conclusions?

Reviewer #1: Yes

Reviewer #2: Yes

3. Has the statistical analysis been performed appropriately and rigorously? 

Reviewer #1: Yes

Reviewer #2: Yes

4. Have the authors made all data underlying the findings in their manuscript fully available?

Reviewer #1: Yes

Reviewer #2: Yes

5. Is the manuscript presented in an intelligible fashion and written in standard English?

Reviewer #1: Yes

Reviewer #2: Yes

6. Review Comments to the Author

Reviewer #1: (No Response)

Reviewer #2: (No Response)

7. PLOS authors have the option to publish the peer review history of their article (what does this mean?). If published, this will include your full peer review and any attached files.

Reviewer #1: **Yes: **Prof.I.Saravanan

Reviewer #2: **Yes: **vishal gangadhar naranje

---

## [Editor Report · Acceptance letter]

28 Nov 2024

PONE-D-24-34684R1 

PLOS ONE

Dear Dr. Salunkhe, 

I'm pleased to inform you that your manuscript has been deemed suitable for publication in PLOS ONE. Congratulations! Your manuscript is now being handed over to our production team.

Kind regards, 

on behalf of

Dr. Himadri Majumder 

Academic Editor

PLOS ONE